# Implicitly regularized interaction between SGD and the loss landscape geometry

## Abstract

We study unstable dynamics of stochastic gradient descent (SGD) and its impact on generalization in neural networks. We find that SGD induces an implicit regularization on the interaction between the gradient distribution and the loss landscape geometry. Moreover, based on the analysis of a concentration measure of the batch gradient, we propose a more accurate scaling rule, Linear and Saturation Scaling Rule (LSSR), between batch size and learning rate.

## 1 Introduction

SGD plays an important role in the success of deep learning. However, we still do not fully understand how SGD works from the perspectives of both optimization behavior and generalization performance. To be specific, SGD is a stochastic approximation of full-batch gradient descent (GD), but SGD generally yields better generalization with a small batch size [27, 23]. Moreover, GD is a discretization of gradient flow (GF) with a finite learning rate, i.e., GF is a GD in the limit of vanishing learning rate, but GD generally performs better with a large learning rate [2, 32, 28, 43]. There are some *scaling rules* [25, 10, 15, 45, 54] on how to tune the learning rate for varying batch sizes, but they fail when the batch size gets large [42, 38, 57, 43, 33]. Especially for a greater data-parallelism to accelerate the training process, we require a more accurate scaling rule for the large-batch regime.

There has been many studies to understand the SGD dynamics and its impacts on generalization in deep neural networks. While they provide some useful and intuitive explanations to help us understand these properties of SGD, unfortunately, some results often rely on impractical assumptions or only apply to a certain range of learning rates and batch sizes. For example, some approximate SGD as a stochastic differential equation (SDE) in the limit of vanishing learning rate [34, 35, 29, 16, 30, 31, 18, 44, 4]. Therefore, in a practical finite learning rate regime, this may not properly describe the SGD dynamics. Moreover, Yaida [52] raises some theoretical issues about the SDE approximation and Li et al. [33] theoretically analyze a sufficient condition for the SDE approximation to fail.

In this paper, we aim to understand the dynamics and the implicit bias of SGD through the analysis of the *interaction* between SGD and the loss landscape of a neural network with minimal assumptions. To be specific, we investigate the unstable dynamics of SGD "at the edge of stability" [6] (Section 4.1-4.2). This investigation leads to a more refined characterization of the edge of stability by the *interaction-aware sharpness* which extends the previous findings for full-batch GD to a general SGD. Then, we introduce a *concentration measure* of the the batch gradient distribution of SGD. By doing so, we find that SGD implicitly regularizes the interaction-aware sharpness and its regularization effect is controlled by the ratio of the concentration measure to learning rate (Section 5.1). Finally, we propose a more accurate scaling rule between batch size and learning rate, based on a novel

Submitted to 36th Conference on Neural Information Processing Systems (NeurIPS 2022). Do not distribute.

34  analysis of the implicit regularization and the concentration measure (Section 5.2). This can be
35  applied to any batch size including the large-batch regime where the previous scaling rules fail
36  [18, 38, 57, 42, 43, 46]. We name it *Linear and Saturation Scaling Rule* (LSSR).

## 2   Stochastic Gradient and Loss Landscape

38  In this section, we review some concepts required for further discussion. We also summarize the
39  notations in Appendix A for a quick reference. We often omit the dependence on some variables and
40  the subscript of the expectation operation when clear from the context.

41  For a learning task, we use a parameterized model (neural network) with model parameter $\theta \in$
42  $\Theta \subset \mathbb{R}^m$. Then we train the model using training data $\mathcal{D} = \{x_i\}_{i=1}^n$ and a loss function $\ell(x; \theta)$.
43  We denote the (total) training loss by $L(\theta) \equiv \frac{1}{n} \sum_{i=1}^n \ell(x_i; \theta)$ for training data $\mathcal{D}$. At time step
44  $t$, we update the parameter $\theta_t$ using GD: $\theta_{t+1} = \theta_t - \eta \nabla_\theta L(\theta_t)$ with a learning rate $\eta$, or using
45  SGD: $\theta_{t+1} = \theta_t - \eta g_b(\theta_t)$ with a mini-batch gradient $g_b(\theta_t) \equiv \frac{1}{b} \sum_{x \in \mathcal{B}_t} \nabla_\theta \ell(x; \theta_t) \in \mathbb{R}^m$ for a
46  mini-batch $\mathcal{B}_t \subset \mathcal{D}$ of size $b$ ($1 \leq b \leq n$).

47  Now, we are ready to introduce some important matrices, $C_b, S_b$, and $H$. First, we define the
48  covariance $C_b(\theta) \equiv \mathrm{Var}[g_b(\theta)] = \mathbb{E}\left[ (g_b(\theta) - \mathbb{E}[g_b(\theta)]) (g_b(\theta) - \mathbb{E}[g_b(\theta)])^\top \right] \in \mathbb{R}^{m \times m}$ and the
49  second moment $S_b(\theta) \equiv \mathbb{E}[g_b(\theta) g_b(\theta)^\top] \in \mathbb{R}^{m \times m}$ of the mini-batch gradient $g_b(\theta)$ over batch
50  sampling for a batch size $1 \leq b \leq n$.[1] The covariance $C_b$ and the second moment $S_b$ satisfy not only
51  $C_b = S_b - S_n$ but also the following equation [15, 29, 49]:

$$C_b = \frac{\gamma_{n,b}}{b}(S_1 - S_n) = \frac{\gamma_{n,b}}{b} C_1, \tag{1}$$

52  where $\gamma_{n,b} = \frac{n-b}{n-1}$ for sampling *without* replacement and $\gamma_{n,b} = 1$ for sampling *with* replacement.
53  We provide a self-contained proof of (1) in Appendix B.1. We note that, for sampling without
54  replacement, many previous works approximate $\gamma_{n,b} \approx 1$ assuming $b \ll n$ [18, 15, 46], but we
55  consider the whole range of $1 \leq b \leq n$ ($0 \leq \gamma_{n,b} \leq 1$ with $\gamma_{n,1} = 1$ and $\gamma_{n,n} = 0$). Second,
56  we define the Hessian $H(\theta) = \nabla_\theta^2 L(\theta) = \mathbb{E}_{x \sim \mathcal{D}}[\nabla_\theta^2 \ell(x; \theta)] \in \mathbb{R}^{m \times m}$ and the operator norm (the
57  top eigenvalue) $\|H\| \equiv \sup_{\|u\|=1} \|Hu\|$ of $H$. We also denote the $i$-th largest eigenvalue and its
58  corresponding normalized eigenvector by $\lambda_i \in \mathbb{R}$ and $q_i \in \mathbb{R}^m$, respectively, for $i = 1, \cdots, m$.

59  Therefore, with these matrices, we can write one of our goals as follows:

60  *We aim to understand how the gradient distribution ($C_b$ and $S_b$) and the loss landscape geometry*
61  *($H$) interact with each other during SGD training.*

62  We investigate this "interaction" in terms of matrix multiplication $HS_b$. To be specific, we consider
63  the trace $\mathrm{tr}(HS_b)$ or its normalized one $\frac{\mathrm{tr}(HS_b)}{\mathrm{tr}(S_b)}$ (will be denoted by $\|H\|_{S_b}$ in Definition 2 later).

## 3   Related Work

65  Some studies investigate the interaction between the gradient distribution and the loss landscape
66  geometry represented by $\mathrm{tr}(HS_b)$ in the context of escaping efficiency [58, Section 3.1], stationarity
67  [52, Section 2.2], and convergence [48, Section 3.1.1]. However, they require some additional
68  assumptions like SDE approximation of SGD [58], the existence of a stationary-state distribution
69  of the model parameter [52, Section 2.3.4], and strong convexity of the training loss function [48],
70  respectively. In this paper, we provide a new insight into the interaction $\mathrm{tr}(HS_b)$ without these
71  assumptions.

72  Convergence of full-batch GD ($b = n$) has been instead analyzed with an upper bound on the
73  interaction $\mathrm{tr}(HS_n)$ with further assumptions for the stable optimization, such as $\beta$-smoothness of

---

[1]These two matrices $C_b$ and $S_b$ are also called the second *central* and *non-central* moments, respectively.
But to avoid confusion, we use the term "second moment" only for the non-central $S_b$.

the objective and $0 < \eta < \frac{2}{\beta}$ (e.g., $\eta = \frac{1}{\beta}$) [39, 41, 37, 3]. [2] However, it may lose useful information of the interaction between $H$ and $S_n$. Moreover, when we train a standard neural network with GD in practice, $\|H\|(\leq \beta)$ increases in the early phase of training and the iterate enters the regime called the edge of stability [6] where $\|H\| \gtrsim \frac{2}{\eta}$, i.e., $\eta \gtrsim \frac{2}{\|H\|} \geq \frac{2}{\beta}$. This contradicts with the assumption for stable optimization and the iterate exhibits unstable behavior with a non-monotonically decreasing loss [51, 50, 6]. We further extend this discussion of unstable dynamics for GD to the case of SGD.

From the generalization perspective, many studies focus on the implicit bias of SGD toward a better generalization [40, 56, 47, 20, 21, 1, 46]. There are mainly two factors known to correlate with the generalization performance: the batch gradient distribution during training [15, 18, 44, 58] and the sharpness of the loss landscape at the minimum [14, 23, 8, 22, 9, 26]. We provide a link between the batch gradient distribution and the sharpness that the model is implicitly regularized to have a low sharpness when the second moment of the batch gradient is large (see Section 5.1).

# 4 Optimization through Loss Landscape

We start by investigating the optimization behavior of SGD through the interaction between SGD and the loss landscape *without* the stochastic differential equation (SDE) approximation.

## 4.1 Unstable Optimization

Using the second-order Taylor expansion, the change in total training loss $L_t = L(\theta_t)$ as the SGD iterate moves from $\theta_t$ to $\theta_{t+1}$ at time step $t$ can be expressed as follows:

$$L_{t+1} - L_t = -\eta \nabla L^\top g_b + \frac{\eta^2}{2} g_b^\top H g_b + O(\|\delta_t\|^3), \tag{2}$$

where $\delta_t = \theta_{t+1} - \theta_t = -\eta g_b$. Thus, we obtain the expected loss difference as follows:

$$\mathbb{E}[L_{t+1}] - L_t = -\eta \nabla L^\top \mathbb{E}[g_b] + \frac{\eta^2}{2} \mathbb{E}[g_b^\top H g_b] + \epsilon \tag{3}$$

$$= -\eta \|\nabla L\|^2 + \frac{\eta^2}{2} \operatorname{tr}\left(\mathbb{E}[H g_b g_b^\top]\right) + \epsilon \tag{4}$$

$$= -\eta \operatorname{tr}(S_n) + \frac{\eta^2}{2} \operatorname{tr}(H S_b) + \epsilon \tag{5}$$

$$= \frac{\eta^2}{2} \operatorname{tr}(S_n) \left[\frac{\operatorname{tr}(H S_b)}{\operatorname{tr}(S_n)} - \frac{2}{\eta}\right] + \epsilon, \tag{6}$$

where $\epsilon = O(\mathbb{E}[\|\delta_t\|^3])$ and $\mathbb{E}[g_b] = \nabla L$ is used. For the moment, we make a minimal assumption that the training loss is locally quadratic, i.e., $\epsilon = 0$ near $\theta_t$, but we will revisit this assumption later (see Section 4.2). Then, the expected loss increases when the following *instability condition* is met:

**Definition 1** (Instability Condition).

$$\frac{\operatorname{tr}(H S_b)}{\operatorname{tr}(S_n)} > \frac{2}{\eta}. \tag{7}$$

We also define *unstable regime* $\mathbb{U} = \{\theta \in \Theta : \frac{\operatorname{tr}(H S_b)}{\operatorname{tr}(S_n)} > \frac{2}{\eta}\}$ and *stable regime* $\mathbb{S} \equiv \mathbb{U}^c$. For a standard non-quadratic loss function, we will show in the following sections that the iterate tends not to stay within the unstable regime $\mathbb{U}$ and operates near at the boundary $\partial \mathbb{S}$ of the stable regime $\mathbb{S}$, called the edge of stability [6]. Cohen et al. [6] mark the edge of stability with $\{\theta \in \Theta : \|H\| = \frac{2}{\eta}\}$ for GD, but we mark with $\partial \mathbb{S} = \{\theta \in \Theta : \frac{\operatorname{tr}(H S_b)}{\operatorname{tr}(S_n)} = \frac{2}{\eta}\}$ for both SGD and GD which provides a more clear and generalized indication as shown in Figure 4 later. On the other hand, for a globally quadratic loss, when the GD iterate satisfies the instability condition, it diverges within the unstable regime [6]. We emphasize that many studies on the convergence of GD usually consider the optimization within

---

[2] $L(\theta_{t+1}) - L(\theta_t) \leq \nabla L^\top (\theta_{t+1} - \theta_t) + \frac{\beta}{2} \|\theta_{t+1} - \theta_t\|^2 = -\eta \|\nabla L\|^2 + \frac{\beta \eta^2}{2} \|\nabla L\|^2 = -\eta(1 - \frac{\beta \eta}{2}) \|\nabla L\|^2$ and thus the loss monotonically decreases when $0 < \eta < \frac{2}{\beta}$.

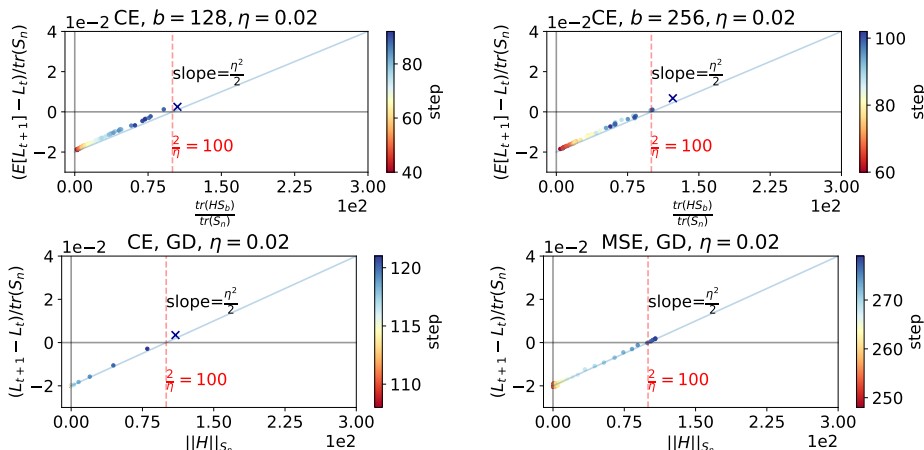

Figure 1: **[An empirical validation of (6) for SGD (top) and (9) for GD (bottom)]** In the early phase, until the iterate enters the edge of stability, it validates (6) and (9) with the blue line with the slope $\frac{\eta^2}{2}$ and x-intercept $\frac{2}{\eta}$. For GD (bottom), they are plotted *after* $\|H\|$ exceeds $\frac{2}{\eta}$ after which $\|H\|_{S_n}$ starts to increase from 0 to $\frac{2}{\eta}$ in a few steps. For cross-entropy loss, we mark the end point with 'x' when the iterate enters the unstable regime. We train 6CNN with $\eta = 0.02$.

the stable regime [39, 41, 37, 3], but GD mostly occurs at the edge of stability after a few steps of training. We will argue that this behavior is crucial for generalization in neural networks.

For later use, we also define the *interaction-aware sharpness* as follows:

**Definition 2** (interaction-aware sharpness)**.**

$$\|H\|_{S_b} \equiv \frac{\text{tr}(HS_b)}{\text{tr}(S_b)}. \tag{8}$$

Here, $\text{tr}(HS_b) \leq \|H\| \text{tr}(S_b)$, i.e., $\|H\|_{S_b} \leq \|H\|$, and the equality holds only when every $g_b$ is aligned in the direction of the top eigenvector of $H$.

Figure 1 (top row) empirically validates (6), showing the normalized loss difference $\frac{\mathbb{E}[L_{t+1}] - L_t}{\text{tr}(S_n)}$ against $\frac{\text{tr}(HS_b)}{\text{tr}(S_n)}$ in the early phase of training before entering the unstable regime. This result implies that the training loss $L(\theta)$ is approximately locally quadratic, i.e., $\epsilon \approx 0$, in the early phase. Especially, for full-batch GD ($b = n$), the instability condition can be rewritten as $\|H\|_{S_n} > \frac{2}{\eta}$ and we have the following relationship between the loss difference $L_{t+1} - L_t$ and $\|H\|_{S_n}$ from (6):

$$L_{t+1} - L_t = \frac{\eta^2}{2} \text{tr}(S_n) \left( \|H\|_{S_n} - \frac{2}{\eta} \right) + \epsilon. \tag{9}$$

Figure 1 (bottom row) shows $\|H\|_{S_n}$ soars from 0 in a few steps after $\|H\|$ exceeds $\frac{2}{\eta}$ [6], satisfying (9) approximately with $\epsilon \approx 0$, before the iterate enters the edge of stability. This result is consistent with the following Proposition for a quadratic training loss $L$. The proof is deferred to Appendix B.2.

**Proposition 4.1.** *For GD with a quadratic $L$, if $\|H\| > \frac{2}{\eta}$ and $0 < \lambda_i < \frac{2}{\eta}$ for all $i \neq 1$, then $|\cos(q_1, \nabla L(\theta_t))|$, $|q_1^\top \nabla L(\theta_t)|$ and $\|H\|_{S_n}$ increase to $1, \infty$ and $\|H\|$, respectively, as $t \to \infty$.*

## 4.2 Non-quadraticity, Asymmetric Valleys and the Edge of Stability

In the previous section, we have shown that the training loss is approximately locally quadratic *before* the iterate enters the edge of stability. However, *after* the iterate enters the edge of stability, i.e., $\frac{\text{tr}(HS_b)}{\text{tr}(S_n)}$ reaches and exceeds $\frac{2}{\eta}$, the step size is relatively large for the sharp loss landscape so that the iterate jumps across the valley [19], and the higher-order terms $\epsilon$ in (6) and (9) become non-negligible and cause a different behavior of the iterate than in the stable regime.

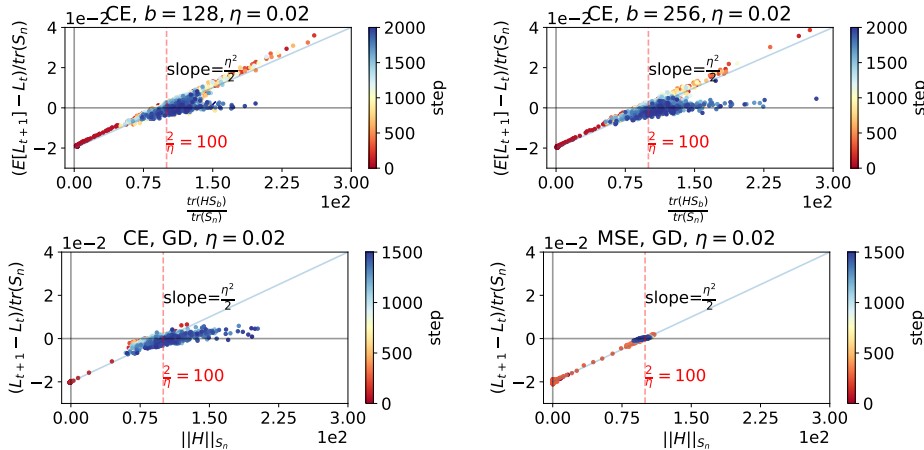

Figure 2: **[Non-quadraticity and overestimation]** The normalized loss difference $\frac{\mathbb{E}[L_{t+1}]-L_t}{\mathrm{tr}(S_n)}$ against $\frac{\mathrm{tr}(HS_b)}{\mathrm{tr}(S_n)}$ during training. After the iterate enters the edge of stability, it often shows a more gentle slope than $\frac{\eta^2}{2}$, especially in the unstable regime.

Figure 2 shows empirical evidences for the *non-quadraticity*. After the SGD/GD iterate enters the edge of stability, when the instability condition $\frac{\mathrm{tr}(HS_b)}{\mathrm{tr}(S_n)} > \frac{2}{\eta}$ is met, the normalized increase in the loss $\left|\frac{\mathbb{E}[L_{t+1}]-L_t}{\mathrm{tr}(S_n)}\right|$ is often smaller than $\frac{\eta^2}{2}\left|\frac{\mathrm{tr}(HS_b)}{\mathrm{tr}(S_n)} - \frac{2}{\eta}\right|$ from (6) and (9) (blue line) when assuming a locally quadratic function. This results in a gentle slope less than $\frac{\eta^2}{2}$.

We hypothesise that due to this non-quadraticity of the training loss, the iterate is discouraged from staying within the unstable regime. Figure 3 demonstrates the asymmetric valley [12] that one side is sharp and the other is flat. In Figure 3 (left), we evaluate the directional sharpness $\|H_\alpha\|_{S_n}$ along the gradient descent direction $-\eta\nabla L(\theta)$ where $H_\alpha \equiv H(\theta - \alpha\eta\nabla L(\theta))$ for $\alpha \in \frac{1}{4} \times [1,2,3,4,5]$, and compare $\|H_\alpha\|_{S_n(\theta)}$ with $\|H\|_{S_n(\theta)}$. At the sharp side, it has a high $\|H\|_{S_n} > \frac{2}{\eta}$ (blue) with the gradient $\nabla L$ and the top eigenvector $q_1(H)$ of the Hessian being highly aligned (cf. Prop. 4.1). However, when the loss landscape gets far from being quadratic, the Hessian and its top eigenvector can change abruptly, $q_1(H_\alpha)$ would not always be aligned with $q_1(H)$ and $\nabla L(\theta)$, and $\|H_\alpha\|_{S_n}$ tends to decrease. This would be a possible explanation for the tendency of decreasing and then oscillating $\|H\|_{S_n}$. See Appendix C.3 for detailed empirical evidences of the above arguments. Figure 3 (right) similarly shows that when the iterate is at a sharp side of the valley, it tends to jump to the other side of a flatter area, and vice versa.

To summarize, we make the following observations for GD in order: (i) $\|H\|$ increases in the beginning (the *progressive sharpening* [6]), (ii) $\|H\|$ exceeds $\frac{2}{\eta}$, (iii) the gradient $\nabla L$ becomes more aligned with the top eigenvector $q_1(H)$ in a few steps, (iv) $\|H\|_{S_n}$ reaches the threshold $\frac{2}{\eta}$ and the iterate jumps across the valley, (v) $\|H\|_{S_n}$ tends to decrease due to the non-quadraticity, and it repeats this process, while $\|H\|_{S_n}$ oscillating around $\frac{2}{\eta}$. We observe a similar behavior with oscillating $\frac{\mathrm{tr}(HS_b)}{\mathrm{tr}(S_n)}$ around $\frac{2}{\eta}$ for SGD. It requires further investigation into the exact underlying mechanisms and we leave it as a future work.

**Remark** (Experiments in Section 4). *We report the experimental results using vanilla SGD/GD without momentum and weight decay, constant learning rate, and no data augmentation. We train a simple 6-layer CNN (6CNN, $m = 0.51M$) on CIFAR-10-8k where* DATASET-$n$ *denotes a subset of* DATASET *with $|\mathcal{D}| = n$ and $k=2^{10} = 1024$. See Appendix C.1-C.3 for the results from other datasets, learning rates and networks (ResNet-9 with $m = 2.3M$ [13] and WRN-28-2 with $m = 36M$ [55]).*

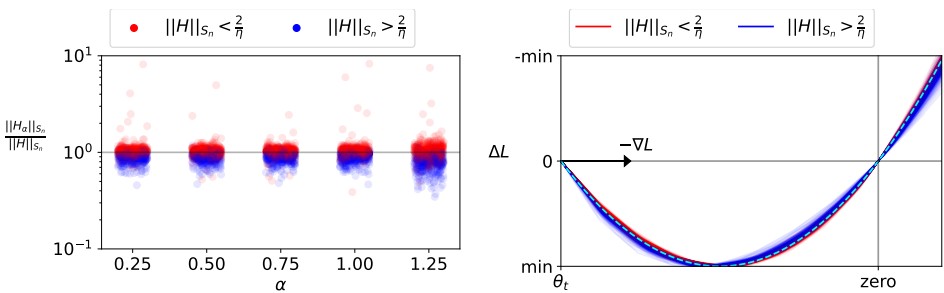

Figure 3: **[Asymmetric valleys]** Left: The ratio $\frac{\|H_\alpha\|_{S_n}}{\|H\|_{S_n}}$ where $H_\alpha = H(\theta - \alpha\eta\nabla L(\theta))$ for $\alpha = \frac{1}{4} \times [1, 2, 3, 4, 5]$ for each $t$ during training. When $\|H\|_{S_n} < \frac{2}{\eta}$ (red), $\|H_\alpha\|_{S_n}$ is usually larger than $\|H\|_{S_n}$. On the other hand, when $\|H\|_{S_n} > \frac{2}{\eta}$ (blue), $\|H_\alpha\|_{S_n}$ is usually smaller than $\|H\|_{S_n}$. Right: The training loss difference along the gradient descent direction, for each $\theta_t$. Each plot is normalized and translated to have the same minimum value and the same zero where $\Delta L = 0$. We also plot the quadratic baseline (cyan dashed curve). When $\|H\|_{S_n} < \frac{2}{\eta}$ (red), it usually becomes sharper across the valley (right-shifted). On the other hand, when $\|H\|_{S_n} > \frac{2}{\eta}$ (blue), it usually becomes flatter across the valley (left-shifted). We train 6CNN using GD with $\eta = 0.04$.

## 5 Generalization through Implicit Regularization

In the previous section, we have empirically demonstrated that the SGD iterate is implicitly discouraged from staying within the unstable regime. Now, we are ready to further analyze this property from the regularization perspective.

### 5.1 Implicit Interaction Regularization (IIR)

First, to understand the effect of batch size $b$ on the gradient distribution, we define the following $\rho_b$:

**Definition 3** (a concentration measure of the batch gradient). We define $\rho_b$ as the ratio of the squared norm of the total gradient $\|\nabla L\|^2$ to the expected squared norm of the batch gradients $\mathbb{E}[\|g_b\|^2]$, i.e.,

$$\rho_b \equiv \frac{\|\nabla L\|^2}{\mathbb{E}[\|g_b\|^2]} = \frac{\text{tr}(S_n)}{\text{tr}(S_b)}. \tag{10}$$

Here, we can write $\|\nabla L\|^2 = \|\mathbb{E}[g_b]\|^2$ and thus the ratio $\rho_b = \frac{\|\mathbb{E}[g_b]\|^2}{\mathbb{E}[\|g_b\|^2]} \leq 1$ is similar to the square of the mean resultant length $\bar{R}_b^2 \equiv \|\mathbb{E}[\frac{g_b}{\|g_b\|}]\|^2 \leq 1$ of the batch gradient $g_b$ [36], especially when $\text{std}[\|g_b\|]$ is small compared to $\mathbb{E}[\|g_b\|]$ (see Appendix C.5 for empirical evidences). Both $\rho_b$ and $\bar{R}_b^2$ are concentration measures and have lower values when the batch gradients $g_b$ are more scattered. Therefore, it is natural to expect that the ratio $\rho_b$ is small for a small batch size $b$, and we will revisit this in more detail in the following section (cf. (12)). We also note that $\rho_n = \bar{R}_n^2 = 1$.

Now, we can rewrite the instability condition $\frac{\text{tr}(HS_b)}{\text{tr}(S_n)} > \frac{2}{\eta}$ (multiplying both sides by $\rho_b$) as $\|H\|_{S_b} > \frac{2\rho_b}{\eta}$. In other words, the interaction-aware sharpness $\|H\|_{S_b}$ is implicitly regularized to be less than $\frac{2\rho_b}{\eta}$. We name this *Implicit Interaction Regularization (IIR)*.

**Definition 4** (Implicit Interaction Regularization (IIR)).

$$\|H\|_{S_b} \leq \frac{2\rho_b}{\eta}. \tag{11}$$

We argue that the upper constraint $\frac{2\rho_b}{\eta}$ in IIR is crucial in determining the generalization performance. With a low constraint, SGD strongly regularizes the interaction-aware sharpness $\|H\|_{S_b}$. We also note that IIR affects not only the magnitude $\|H\|$ but also the *directional* interaction. In other words, IIR discourages the batch gradients from aligning with the top eigensubspace of the Hessian that is spanned by a few largest eigenvectors of the Hessian (cf. [11]).

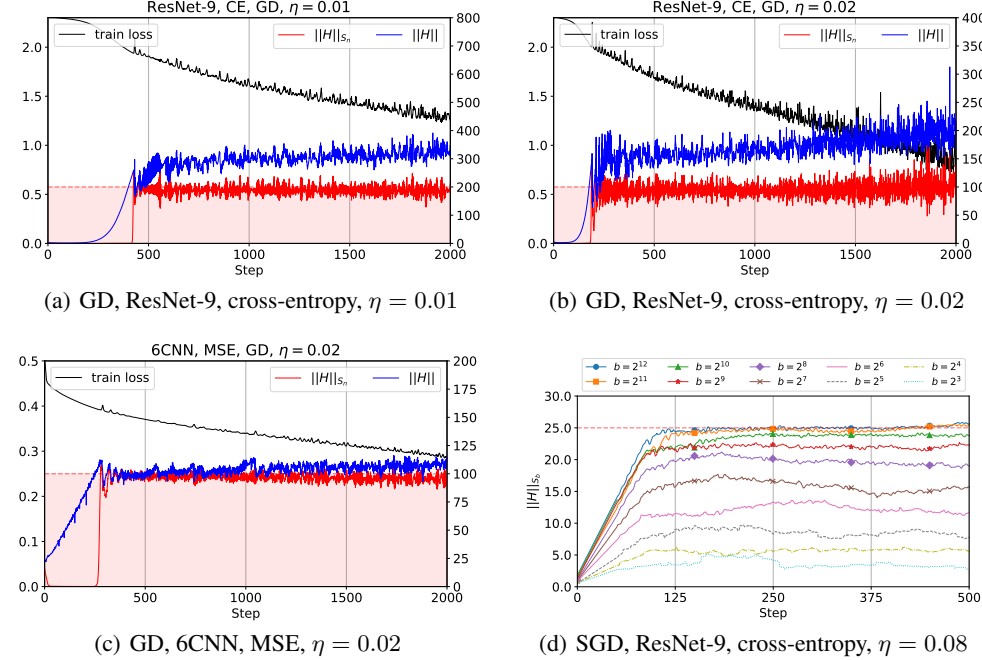

(a) GD, ResNet-9, cross-entropy, $\eta = 0.01$

(b) GD, ResNet-9, cross-entropy, $\eta = 0.02$

(c) GD, 6CNN, MSE, $\eta = 0.02$

(d) SGD, ResNet-9, cross-entropy, $\eta = 0.08$

Figure 4: **[A clear indication of the edge of stability]** (a)-(c): After a few steps of full-batch training, $\|H\|$ (blue) hovers **above** $\frac{2}{\eta}$ [6], but $\|H\|_{S_n}$ (red, defined in (8)) oscillates **around** $\frac{2}{\eta}$ (red dashed horizontal line). The edge of stability is more evident in the latter (red). Curves are plotted for every step. We train a model on CIFAR-10-8k ($n = 2^{13}$) using (a)/(b) cross-entropy loss with $\eta = 0.01/0.02$, respectively, and (c) MSE with $\eta = 0.02$. (d): We plot curves $\|H\|_{S_b}$ when trained with various $b$'s. After a few steps (around 125), they reach the threshold which linearly increases as $b$ becomes larger when $b \ll n = 2^{13}$, and saturates to $\frac{2\rho_b}{\eta} \approx \frac{2}{\eta}$ when $b$ is large. Curves are smoothed for visual clarity. We use SGD with $b \in \{2^3, \cdots, 2^{12}\}$ and $\eta = 0.08$.

Figures 4(a)-4(c) show that, for GD ($\rho_n = 1$), the interaction-aware sharpness $\|H\|_{S_n}$ (red) oscillates *around* $\frac{2}{\eta}$ and exhibits IIR. This result is consistent with Cohen et al. [6] that $\|H\|$ hovers *above* $\frac{2}{\eta}$ for GD. This is because, as mentioned earlier, $\frac{2}{\eta} \approx \|H\|_{S_n} \leq \|H\|$ and the equality holds only when the gradient $\nabla L$ and the top eigenvector $q_1$ of $H$ are aligned, but generally they are not. For this reason, IIR provides a tighter relation and more clearly identifies the edge of stability than Cohen et al. [6]. These results are also consistent with Prop. 4.1 that $\|H\|_{S_n}$ suddenly increases from 0 to $\frac{2}{\eta}$ in a few steps after $\|H\|$ exceeds $\frac{2}{\eta}$ (see Appendix C.3-C.4 for more). Moreover, IIR also applies to a general SGD training with $1 \leq b \leq n$. Figure 4(d) shows IIR for SGD with different batch sizes $b \in \{2^3, \cdots, 2^{12}\}$. The upper bound ($2\rho_b/\eta$ according to (11)) of $\|H\|_{S_b}$ is higher when using a larger batch size, but limited to less than $2/\eta$ ($\rho_b \leq 1$). We will further discuss this behavior with an investigation of $\rho_b$ in the following section.

## 5.2 Linear and Saturation Scaling Rule (LSSR)

The ratio $b/\eta$ of batch size $b$ to learning rate $\eta$ has long been believed as an important factor influencing the generalization performance, and the test accuracy has observed to be similar when trained with the same ratio $b/\eta = b'/\eta'$, i.e., $b' = kb$ and $\eta' = k\eta$ for $k > 0$. This is called the linear scaling rule (LSR) [25, 10, 18, 44, 57]. They argue that LSR holds because $\theta_{t+k} - \theta_t = -\frac{\eta}{b} \sum_{i=0}^{k-1} \sum_{x \in \mathcal{B}_{t+i}} \nabla \ell(x; \theta_{t+i}) \approx -\frac{\eta}{b} \sum_{i=0}^{k-1} \sum_{x \in \mathcal{B}_{t+i}} \nabla \ell(x; \theta_t) = -\frac{\eta'}{b'} \sum_{x \in \mathcal{B}_{t:t+k}} \nabla \ell(x; \theta_t)$ assuming $\nabla \ell(\theta_{t+i}) \approx \nabla \ell(\theta_t)$ for $0 \leq i < k$, where $\mathcal{B}_{t:t+k} \equiv \cup_{i=0}^{k-1} \mathcal{B}_{t+i}$ and $|\mathcal{B}_{t:t+k}| = kb = b'$. However, the assumption is false and the gradient oscillates mostly with a negative cosine value $\cos(g_b(\theta_t), g_b(\theta_{t+1})) < 0$ between two consecutive gradients after entering the edge of stability

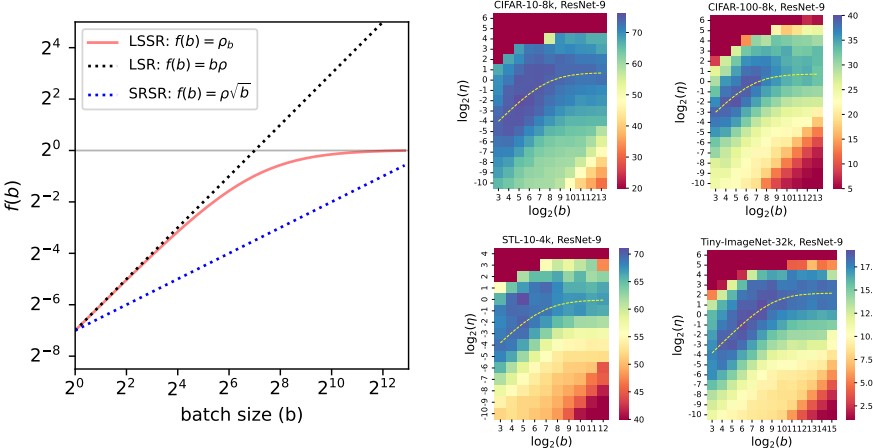

Figure 5: **[Linear and Saturation Scaling Rule (LSSR)]** Left: LSSR (red) in (12), LSR (black dotted line) [10] and SRSR (blue dotted line) [15]. For LSSR, we can observe both linear and saturation regions ($n = 8k, \rho = 2^{-7}$). Right: Heatmaps of test accuracy for models trained with a large number of pairs of $(b, \eta)$ on CIFAR-10-8k , CIFAR-100-8k , STL-10-4k, and Tiny-ImageNet-32k (from left to right, from top to bottom). It does not follow either LSR or SRSR, but LSSR. We also plot $f(b) = \rho_b$ (yellow dashed curve) for some $\rho$ on each heatmap. Note that they are all log-log plots and thus a slope of 1 means it is linear.

195  (see Appendix C.3). Moreover, LSR fails when the batch size is large [18, 38, 57, 43, 46]. On the
196  other hand, Krizhevsky [25], Hoffer et al. [15] propose the square root scaling rule (SRSR) with
197  another ratio $\sqrt{b}/\eta$ to keep the covariance of the parameter update constant for $b \ll n$ based on
198  $\mathrm{Var}[\eta g_b] = \eta^2 C_b = \frac{\gamma_{n,b}\eta^2}{b}C_1 \approx \frac{\eta^2}{b}C_1$. However, Shallue et al. [42] show that both LSR and SRSR
199  do not hold in general.

200  Based on the analysis of IIR with a new ratio $2\rho_b/\eta$ in the previous section, we explore why LSR fails
201  in the large-batch regime and provide a more accurate rule to explain the generalization performance
202  of the models trained with various choices of batch size and learning rate pairs $(b, \eta)$.

203  To this end, we investigate the concentration measure $\rho_b = \mathrm{tr}(S_n)/\mathrm{tr}(S_b)$. By combining two
204  equations, $C_b = S_b - S_n$ (by definition) and $C_b = \frac{\gamma_{n,b}}{b}(S_1 - S_n)$ in (1), we can obtain $S_b =$
205  $C_b + S_n = \frac{\gamma_{n,b}}{b}S_1 + (1 - \frac{\gamma_{n,b}}{b})S_n$. Therefore, we have $\mathrm{tr}(S_b) = \frac{\gamma_{n,b}}{b}\mathrm{tr}(S_1) + (1 - \frac{\gamma_{n,b}}{b})\mathrm{tr}(S_n)$,
206  which leads to the following equation:

$$\rho_b \equiv \frac{\mathrm{tr}(S_n)}{\mathrm{tr}(S_b)} = \frac{\mathrm{tr}(S_n)}{\frac{\gamma_{n,b}}{b}\mathrm{tr}(S_1) + (1 - \frac{\gamma_{n,b}}{b})\mathrm{tr}(S_n)} = \underbrace{\frac{1}{\frac{\gamma_{n,b}}{b}\frac{1}{\rho} + (1 - \frac{\gamma_{n,b}}{b})}}_{(*)} \approx \begin{cases} \frac{b}{\gamma_{n,b}}\rho \approx b\rho & \text{if } b \text{ is small} \\ 1 & \text{if } b \text{ is large} \end{cases}$$

(12)

207  from (10) where $\rho = \rho_1 = \mathrm{tr}(S_n)/\mathrm{tr}(S_1)$. Note that $\rho$ is (much) smaller than 1 because $\nabla\ell(x_i)$
208  has different direction for each $x_i$ and $\mathrm{tr}(S_n) = \|\nabla L\|^2 = \|\frac{1}{n}\sum_i \nabla\ell(x_i)\|^2 \le \frac{1}{n}\sum_i \|\nabla\ell(x_i)\|^2 =$
209  $\mathrm{tr}(S_1)$. In other words, $1/\rho$ is (much) larger than 1 (see Appendix C.5).

210  Figure 5 (left) demonstrates a new scaling rule with the ratio $\rho_b/\eta$, called the *Linear and Saturation*
211  *Scaling Rule* (LSSR), with the two regimes that (i) $\rho_b$ is almost linear when $b \ll n$ (linear regime) and
212  (ii) $\rho_b$ saturates when $b$ is large (saturation regime), which are also shown in Figure 4(d). It depends
213  on which part of the denominator $(*)$ in (12) dominates the other. First, when $b \ll n$, then $\gamma_{n,b}/b$ is
214  not very small and the first term $\frac{\gamma_{n,b}}{b}\frac{1}{\rho}$ dominates the second term $1 - \frac{\gamma_{n,b}}{b}$ since $\frac{1}{\rho} \gg 1$. Second, as
215  $b$ becomes large, $\gamma_{n,b}/b \approx 0$ and the second term $(\approx 1)$ dominates the first term. Thus, $\rho_b$ saturates
216  to 1 and is not linearly related to $b$, and LSR is no longer valid. The above arguments also hold for
217  the batches sampled *with* replacement where the only modification is $\gamma_{n,b} = 1, \forall b$ in (12). Figure 5
218  (right) empirically supports LSSR with the test accuracies when trained with various combinations of
219  pairs $(b, \eta)$. To be specific, the optimal learning rate is almost linear when $b$ is small, but it saturates

when $b$ is large. We also plot $f(b) = \rho_b$ (the yellow dashed curve) for some $\rho$. Note that Figure 8 of Shallue et al. [42, Section 4.7] shows similar "linear and saturation" behaviors supportive of LSSR on other datasets (see also Figure 7 of Zhang et al. [57, Section 4.3]).

**Remark** (Experiments in Section 5). *We train models using vanilla SGD/GD without momentum and weight decay, constant learning rate, and no data augmentation. For Figure 5, we use subsets of the datasets CIFAR-10 [24], CIFAR-100 [24], STL-10 [5], and Tiny-ImageNet (a subset of ImageNet [7] with $3 \times 64 \times 64$ images and 200 object classes). We use a large number of epochs (800) and batch normalization [17] to achieve a zero training error even with a large $b$ and a small $\eta$. However, in the lower right corner (red area) of each heatmap in Figure 5 (right), when $b$ is too large or $\eta$ is too small so that $\|\theta_{t+1} - \theta_t\| = \eta\|g_b\|$ is too small, it requires an exponentially large number of steps for the iterate to enter the edge of stability. Thus, in this case, the assumption in Goyal et al. [10], $\nabla\ell(\theta_t) \approx \nabla\ell(\theta_{t+i})$ for $0 \le i < k$, approximately holds and the reasoning on LSR is valid. However, this only holds for a non-practical $(b, \eta)$ which shows a suboptimal performance. See Appendix C.4-C.5 for the results from other networks and hyperparameters.*

## 6   Discussion

We provide a new insight on the link between the batch gradient distribution and the sharpness of the loss landscape. In this section, we reconcile our arguments with some previous studies.

Jastrzębski et al. [18] explain the optimization behavior of SGD with the SDE approximation $d\theta_t = -\nabla L(\theta_t)dt + \sqrt{\frac{\eta}{b}}C_1^{1/2}dW(t)$ of the SGD where $W$ is an $m$-dimensional Brownian motion. Therefore, the same ratio $\frac{\eta}{b} = \frac{\eta'}{b'}$ leads to the same SDE, which implies LSR. Moreover, a large $\frac{\eta}{b}$ implies a large diffusion in SDE, which has been linked with the escaping efficiency from a sharp local minimum in Zhu et al. [58]. We instead argue that a large second moment $\text{tr}(S_b)$ (compared to $\text{tr}(S_n)$) and a large $\eta$ lead to a low constraint $2\rho_b/\eta$ on the interaction-aware sharpness. We emphasize that we do not model SGD with SDE and thus our argument is applicable to a practical learning rate regime.

Wu et al. [49] empirically show that what is important for the generalization performance of a neural network is not the class to which the gradient distribution belongs, but the second moment of the distribution. This is consistent with our arguments with the interaction $\text{tr}(HS_b)$ and the concentration measure $\rho_b = \text{tr}(S_n)/\text{tr}(S_b)$, because they depend on the second moment $S_b$, not on the class of the gradient distribution.

Recently, Li et al. [33] suggest a necessary condition that the "noise-to-signal ratio" needs to be large for LSR (and the SDE assumption) to hold. This is consistent with our result on the linear regime (where $b$ and $\rho_b$ are small) because the noise-to-signal ratio is approximately the inverse of the "signal-to-noise" ratio $\rho_b = \text{tr}(S_n)/\text{tr}(S_b)$, but defined for an equilibrium distribution. We provide not only the necessary condition but also the sufficient condition for LSR with a novel scaling rule LSSR applicable to every batch size including where LSR fails (the saturation regime).

## 7   Conclusion

From an analysis of unstable dynamics of SGD (Section 4.1) and the instability condition (Definition 1), we clearly mark the edge of stability (Figure 4) with the interaction-aware sharpness $\|H\|_{S_b}$ (Definition 2) and show the presence of the implicit regularization effect on the interaction between the gradient distribution and the loss landscape geometry (IIR) (Section 5.1, Definition 4). Moreover, introducing the concentration measure $\rho_b$ of the batch gradient (Definition 3, (12)), we link the second moment of the gradient distribution and the sharpness of the loss landscape, and propose a new scaling rule called Linear and Saturation Scaling Rule (LSSR) (Section 5.2, Figure 5). Due to the simplicity of the analysis, we hope that our insights will motivate the future work toward understanding various learning tasks.

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
