# OpenReview forum: "Implicitly regularized interaction between SGD and the loss landscape geometry"
_NeurIPS.cc/2022/Conference — NeurIPS 2022 Submitted_

### Official Review · Reviewer_fFDo · 2022-07-09

**Rating:** 7
**Confidence:** 5
**Soundness:** 4 excellent
**Presentation:** 3 good
**Contribution:** 3 good

**Summary:**

This paper makes several contributions:
 1.  The authors propose a new characterization of 'edge of stability' which is said to improve over the original characterization from Cohen et al. (2021) in that it (1) fits full-batch GD better, and (2) generalizes to SGD.  Specifically, whereas the original characterization is that the sharpness (maximum Hessian eigenvalue) hovers **just above** the value $2/\eta$, the authors propose characterizing the EoS as a regime in which the expected loss difference that is predicted by a quadratic taylor approximation hovers **around** the value zero (here, the expectation is wrt randomness in minibatch sampling).  For SGD, this condition comes out to $\frac{\text{tr}(H S_b)}{\|g\|^2} \approx \frac{2}{\eta}$, where g is the (full-batch) gradient, H is the Hessian, and $S_b$ is the second moment matrix of minibatch gradients.  For full-batch GD, the condition reduces to $\frac{g^T H g}{\|g\|^2} \approx \frac{2}{\eta}$.  In the case of full-batch GD, the authors argue that their condition $\frac{g^T H g}{\|g\|^2} \approx \frac{2}{\eta}$ improves over the original condition $\|H\|_2 \gtrapprox 2/\eta $ in that  $\frac{g^T H g}{\|g\|^2}$ empirically hovers right around $2/\eta$ (Figure 4), whereas there is usually a gap between $\|H\|_2$ and  $2/\eta $.  More importantly, it is currently an open question how the 'edge of stability' phenomenon generalizes to SGD, and the authors argue that their condition $\frac{\text{tr}(H S_b)}{\|g\|^2} \approx \frac{2}{\eta}$ holds during SGD.

 2.  The authors investigate the behavior of gradient descent at the EoS (Figures 1-3).   They find, essentially, that gradient descent is leaping back and forth across an asymmetric valley (https://arxiv.org/abs/1902.00744).  At every iteration where  $\frac{g^T H g}{\|g\|^2}$ is above $2/\eta$, it dips below $2/\eta$ on the next step.  At every iteration where $\frac{g^T H g}{\|g\|^2}$ is below $2/\eta$, it rises above $2/\eta$ on the next step.  Note that a similar point has been made (with no experiments) in the contemporaneous work of Chen & Bruna (https://arxiv.org/abs/2206.04172).

 3.  The authors propose a new scaling rule that describes how the linear rate ought to be scaled with the batch size in order to preserve the same implicit regularization.  Namely, the authors suggest that two training runs have the same implicit regularization whenever $\text{tr}(S_b) / [\eta \| g\|^2]$ is the same, and they present evidence (Figure 5) in support of their hypothesis.  Their scaling rule reduces to the well-known linear scaling rule in the limit of small batches, and reduces to "no scaling" in the limit of large batches.

**Questions:**

- In Figure 5, where does the yellow dotted line come from?

- The plots of $||H||$ in Figure 31 look weird to me -- typically, the sharpness along the optimization trajectory is smaller when the batch size is smaller.  Could there be an error here?


**Limitations:**

Yes

**Strengths And Weaknesses:**

In what follows, I will reference the contributions I listed above by their numbers.

- Weakness of contribution #1: I could be wrong, but I don't think the new EoS characterization will generalize to momentum gradient descent (either with Polyak-style or Nesterov-style momentum).  In the Cohen et al EoS paper, it is intuitively clear why the sharpness should hover just above $2/\eta$: if the training objective is modeled by its quadratic Taylor approximation, we see that gradient descent cannot linger for long in any region where the sharpness exceeds $2/\eta$, because it will oscillate with exponentially growing magnitude in a certain direction until it leaves the region.  The extensions in that paper to Nesterov and Polyak momentum are based on the same reasoning.  By contrast, the underlying logic behind the new proposed EoS characterization (which is based on the idea that the loss is always non-monotonic) is less clear to me, and one consequence is that I don't see how it can be generalized to momentum.

- Strength of contribution #1: the proposed extension of EoS to SGD is interesting (note that it's basically a quadratic Taylor approximation version of the phenomenon reported in Cohen et al (2021) appendix H).  Nothing has been published in the literature addressing the question of how EoS might generalize to SGD, so any experimental result that seems to hold consistently across a wide range of networks is valuable.

- Strength of contribution #2: I think that these experiments will be useful for people interested in how gradient descent manages to remain semi-stable at the EoS.

- Weakness of contribution #3: both $\text{tr}(S_b)$ and $\|g\|^2$ will vary over the course of training, and I don't see any reason why their ratio should remain constant.  Thus, the scaling rule is not even well-defined.

---

> ### Author Response · Authors · 2022-08-01
> **A3. About the weaknesses and questions**
>
> - Weakness of contribution #1: I could be wrong, but I don't think the new EoS characterization will generalize to momentum gradient descent (either with Polyak-style or Nesterov-style momentum).
>     - There are Proposition 1 (Vanilla GD), Theorem 1 (Polyak) and Theorem 2 (Nesterov) in Cohen et al. [6]. They have the same form of "... If $a>\frac{2}{\eta}f(\beta)$, then the sequence $\tilde x_t = q^\top x_t$ will diverge". Note that they consider the element of $x_t$ along the sharpest direction. Their proof is based on the formulation of a second-order difference equation $\tilde x_{t+1}+p_1\tilde x_{t}+p_2\tilde x_{t-1}=0$ by Theorem 2.37 in Elaydi (2005). To be specific, when the condition $a>\frac{2}{\eta}f(\beta)$ holds, then the sequence diverges because the characteristic equation for the second-order difference equation has a solution with the magnitude $>1$, and it oscillates because $p_1>0$ and $p_2>0$. Therefore, $q^\top \delta_t$ oscillates. Similar to our Proposition 4.1, we can prove the behavior of $|\cos(q,\delta)|\rightarrow 1$ and $\frac{\mathbb{E}[\delta^\top H\delta]}{||\delta||^2}\rightarrow ||H||$ in the limit of $t\rightarrow \infty$. Moreover, beyond the asymptotic analysis of the second-order difference equation in the limit of $t\rightarrow \infty$, we also prove the exponential growth of $|q^\top \delta|$ (similar to eq (34) in Appendix B.2), and due to the exponential growth, it only takes a few steps for $\frac{\mathbb{E}[\delta^\top H\delta]}{||\delta||^2}$ to exceed $\frac{2}{\eta}f(\beta)$.
>     - In case of vanilla GD, this can explain a high instability of $||H||\_{S_n}$ when over $\frac{2}{\eta}$ (tendency to go below $\frac{2}{\eta}$), while $||H||$ keeps increasing even after it exceeds $\frac{2}{\eta}$ and hovers above $\frac{2}{\eta}$.
>  As $||H||_{S_n}$ exceeds $\frac{2}{\eta}$,  Figure 24 shows that $|\cos(q,g)|$ increases to 1 within a few steps, $\cos(g,g')$ decreases to -1, the iterate oscillates along $q$. $|\cos(q,q')|$ drops from 1 to 0, and the shape of basin changes (non-quadraticity).
>
> - Weakness of contribution #3: both tr(Sb) and |g|2 will vary over the course of training, and I don't see any reason why their ratio should remain constant. Thus, the scaling rule is not even well-defined.
>     - As shown in (*) in Eq (12), we are only required to see how $\rho=\frac{||\nabla L||^2}{\mathbb{E}[||\nabla \ell||^2]}$ varies over the course of training. It is not a constant, but it seems to have a similar value regardless of the choice of $\eta$ as shown in Figure 37 ($\frac{1}{\rho}$ is about 100). However, we still need further investigation. Thank you for pointing it out.
> - In Figure 5, where does the yellow dotted line come from?
>     - For each heatmap, we pick $\rho$ and plot $\eta =\eta(b) = \frac{2}{C’}\rho_b =\frac{2}{C’}\frac{1}{\frac{\gamma_{n,b}}{b}\frac{1}{\rho} +(1-\frac{\gamma_{n,b}}{b})}$ as a function of $b$ (yellow dashed curve) for some $C’$ so that IIR has constant effect $||H||_{S_b}\leq C’$ which is approximately the optimal regularization with the best generalization performance. Note that the red curve (LSSR) in Figure 5 (left) and the yellow curves in Figure 5 (right) come from the same equation (12) (without approximation) but different $n$ and $\rho$.
> - The plots of $||H||$ in Figure 31 look weird to me -- typically, the sharpness along the optimization trajectory is smaller when the batch size is smaller. Could there be an error here?
>     - $\mathsf{P}$: The sharpness along the optimization trajectory is smaller when the batch size is smaller.
>     - For $||H||$, Figure 30 and Figure 31 are the same. Figure 30 demonstrates that $\mathsf{P}$ is true in a wide range of $b$. This tendency is relatively less evident when $b$ is large in the saturation regime of $\rho_b$. Keskar et al. [23] have shown that SGD with a small batch size converges to a flat minimum (small sharpness) and yields better generalization. If we misunderstood the question, please correct us.

---

### Official Review · Reviewer_etxE · 2022-07-14

**Rating:** 3
**Confidence:** 3
**Soundness:** 1 poor
**Presentation:** 1 poor
**Contribution:** 2 fair

**Summary:**

The authors explore the notion of edge of stability for SGD dynamics, they introduce some quantities and explain / illustrate why they make sense for the understanding of the dynamics. They then propose a novel "linear and saturation scaline rule" and provide experimental support.

**Questions:**

- eq (3): over what are you taking the expectation ? if it is the full expectation then you should also have $\mathbb{E}[L_t]$, I guess you are taking the conditional expectation but this is not clear.
- line 96: a bit weird to name a set a "regime". It would also be clearer to keep the $\theta$ index in $H$: $H_\theta$, to make clear that the hessian depends on $\theta$.
- unclear experiments in fig 1: what is the precise setting ? what dataset ?
- l110: " This result implies that the training loss L(θ) is approximately locally quadratic, i.e., ε ≈ 0, in the early phase." Why and what does this mean ? isn't any twice differentiable function approximately locally quadratic ?
- "We hypothesise that due to this non-quadraticity of the training loss, the iterate is discouraged from staying within the unstable regime." I don't see why this is due to non-quadraticity. If you want to optimise $f(x, y) = |y| x^2$ for example, which is always "locally quadratic": then for a large step-size, the iterates are "discouraged from staying within the unstable regime" too.
- Figure 3: what is $\Delta L$ ?
- Figure 4: very hard to read labels, especially in fig (d)
- equation 12: "if $b$ is small", "if $b$ is large": how small ? how large ?

**Strengths And Weaknesses:**

Overall I believe there are some interesting ideas and observations. For example Figure 5 (right) (which illustrates the linear and saturation scaling rule) is rather convincing. Some quantities which are introduction seem to make sense. However the paper and the claims are not at all convincing enough. The abstract and introduction are very superficial and it is not even clear what the authors want to prove / show. The related work is neglected.

I am also bothered by some very vague and imprecise claims and sentences. The title to start with: "Implicitly regularized interaction between SGD and the loss landscape geometry". I don't understand what this means: what is a "regularised interaction" ?
Then in the text: "we find that SGD induces an implicit regularization on the interaction between the gradient distribution and the loss landscape geometry.", "we find that SGD implicitly regularizes the interaction-aware sharpness". I do not manage to understand these sentences, they are very vague and the meaning is not clear at all.

The fact that the paper is not well written with very unprecise statements and claims makes me heavily doubt on the sanity and significance of the presented results. In the following questions I give a few examples to support my opinion.

---

> ### Author Response · Authors · 2022-08-01
> **A2-1. About the presentation**
>
> - Implicit Regularization
>     - The loss minimization for an **over-parameterized** model is an **under-determined** problem with multiple global minima (manifold of global minima). Many global minima horribly generalize, and thus many researchers hypothesize that SGD plays an important role in biasing toward a specific set of global minima that generalize well. This effects is called “implicit bias” or “implicit regularization”.
> - Interaction (between the gradient distribution and the loss landscape geometry)
>     - The terminology "interaction" comes from Thomas et al. [48] ("interplay between noise and curvature").
> Many previous work upper bounds (i) $g_b^\top H g_b\leq \beta||g_b||^2 $ under impractical $\beta$-strong convexity assumption (L72-78) or (ii) $ \text{tr}(HS_b)=\mathbb{E}[g_b^\top H g_b] \leq ||H||\mathbb{E}[||g_b||^2]=||H||\text{tr}(S_b)$.
> These upper bounds ignore important features of the interaction between $g_b$ and $H$ in the formulation of the matrix-vector multiplication $Hg_b$ and matrix-matrix multiplication $HS_b$. For example, the upper bounds ignore **how much the gradient $g_b$ is aligned with the sharpest direction**, the top eigenvector of $H$. If they are not aligned, then the upper bounds are loose and uninformative. We keep using the matrix-vector multiplication rather than using the upper bounds to understand the interaction between the gradient distribution and the loss landscape geometry. In other words, we express the interaction between the gradient distribution ($C_b$ and $S_b$) and the loss landscape geometry $H$ in terms of matrix-vector multiplication $Hg_b$ or matrix-matrix multiplication $HS_b$.

---

> ### Author Response · Authors · 2022-08-01
> **A2-2. Others**
>
> - eq (3): over what are you taking the expectation ? if it is the full expectation then you should also have E[L_t], I guess you are taking the conditional expectation but this is not clear.
>     - Yes, it is conditional expectation. We will clarify it. At the current $\theta_t$, we can obtain the expected loss difference over batch sampling $\mathcal{B}\sim \mathcal{D}^b$ as follows: $\mathbb{E}[L(\theta_t -\eta g_b)]-L(\theta_t) = \cdots $
> - line 96: a bit weird to name a set a "regime". It would also be clearer to keep the θ index in H: Hθ, to make clear that the hessian depends on θ.
>     - We use the word regime like Cohen et al. [6] (EOS). $H$ and $S_b$ depend on $\theta$. We often omit the dependence on $\theta$ for simplicity, but to avoid confusion, we will simplify notations sparingly.
> - unclear experiments in fig 1: what is the precise setting ? what dataset ?
>     - We use CIFAR-10-8k for Figure 1. Detailed settings are given in Remark (L148). We update the caption accordingly.
> - l110: " This result implies that the training loss L(θ) is approximately locally quadratic, i.e., ε ≈ 0, in the early phase." Why and what does this mean ? isn't any twice differentiable function approximately locally quadratic ?
>     - We mean by "locally quadratic" only when $\epsilon\approx 0$ for the Taylor expansion **along the direction of $\delta_t=-\eta g_b$** within the range of $\theta=\theta_t +\alpha\delta_t, \alpha\in\[0,1\]$.
>     - Fig 1 and 2 show that $\epsilon$ in eq (6) is negligibly small before the iterate enter the edge of stability, but not in the edge of stability.
> - "We hypothesise that due to this non-quadraticity of the training loss, the iterate is discouraged from staying within the unstable regime." I don't see why this is due to non-quadraticity. If you want to optimise f(x,y)=|y|x2 for example, which is always "locally quadratic": then for a large step-size, the iterates are "discouraged from staying within the unstable regime" too.
>     - Our notion of locally quadraticity depends on $\delta_t$ and thus $f(x,y)=|y|x^2$ is not always "locally quadratic". A global quadratic function is always locally quadratic even for a large step-size.
> - Figure 3: what is $\Delta L$
>     - We will clarify $\Delta L(\theta) = L(\theta)-L(\theta_t)$ and $\Delta L(\theta=\text{zero})=L(\theta=\text{zero})-L(\theta_t)=0$ in the caption.
> - Figure 4: very hard to read labels, especially in fig (d)
>     - We will increase the label size.
> - equation 12: "if b is small", "if b is large": how small ? how large ?
>     - It depends on $\rho=\rho_1$. Approximately, we have $\frac{\gamma_{n,b}}{b}\approx \frac{1}{b}$ (this is more appropriate approximation than $\gamma_{n,b}\approx 1$ since $\gamma_{n,n}=0$).
>  The  relative difference between the two values $\frac{1}{b}\frac{1}{\rho}$ and $1-\frac{1}{b}$ determines the shape of the graph $f(b)=\rho_b$. In other words, the first term becomes dominant when $b$ gets smaller and the second term becomes dominant when $b$ gets larger. When satisfying $\frac{1}{\rho}\approx b$ (assuming $\frac{1}{\rho} \gg 1$), the two terms match each other. This result roughly indicates that $f(b)$ "bends" at $b=\frac{1}{\rho}$.
>     - Figure 36 (right) shows that the batch size where $f(b)$ bends (roughly $b=\frac{1}{\rho}$) becomes larger as $\rho$ becomes smaller in order of blue/orange/green/red ($\rho=2^{-5}/2^{-7}/2^{-9}/2^{-11}$ and $b=2^5/2^7/2^9/2^{11}$, respectively).
>     -
> $$\rho_b=\frac{1}{\frac{\gamma_{n,b}}{b}\frac{1}{\rho}+(1-\frac{\gamma_{n,b}}{b})}=
>     \begin{cases}
> \rho & \text{if }b=1\\\\
> 1& \text{if } b= n
>     \end{cases}$$
>     -
> $$\rho_b=\frac{1}{\frac{\gamma_{n,b}}{b}\frac{1}{\rho}+(1-\frac{\gamma_{n,b}}{b})} = \frac{1}{1+\frac{\gamma_{n,b}}{b}(\frac{1}{\rho}-1)}\approx \frac{b\rho}{b\rho+1}
> \approx\begin{cases}
> b\rho& \text{if $b$ is small, } b\ll n \text{ or } b\rho\ll 1\\\\
>  1& \text{if $b$ is large, } b\approx n \text{ or } b\rho \gg 1
>     \end{cases}$$

---

### Official Review · Reviewer_xxYn · 2022-07-17

**Rating:** 3
**Confidence:** 4
**Soundness:** 2 fair
**Presentation:** 3 good
**Contribution:** 2 fair

**Summary:**

The paper studies the instability in SGD optimization dynamics (edge of stability) and how that impacts generalization. In particular, the authors propose the idea of interaction-aware sharpness, which I believe is original. Based on a definition of implicit interaction regularization, the authors propose the linear and saturation learning rate scaling rule (LSSR). Most of the findings are supported by empirical evidences but theoretical support is at an intuitive level.

**Questions:**

* My feeling is that the current theoretical setup may explain the gap between batch optimization and minibatch optimization but that is not the same as explaining generalization because generalization depends on the model class size. Can you agree or disagre?
* More specifically, do I understand correctly that the theoretical setup does not assume anywhere that the model is a neural network? Then why are all the experiments on neural networks?
* Can we motivate the linear and saturation scaling rule better? I am not sure why keeping the right-hand side of Eq. (11) constant leads to optimal scaling.

**Strengths And Weaknesses:**

Strengths:
 * The idea of interaction-aware sharpness seems to be original.
 * Empirical evidence to support the self-regulation of interaction-aware sharpness around the edge of stability.
 * Intuituive argument for learning rate scaling.

Weaknesses:
 * Learning rate scaling only supported by a vague argument and empirical evidence.
 * Weak connection between the concepts proposed in the paper and generalization.
 * Nothing in the theory related to neural networks.

---

> ### Author Response · Authors · 2022-08-01
> **A1-1. About the Lack of Theoretical Support**
>
> - Most of the findings are supported by empirical evidences but theoretical support is at an intuitive level.
>     - As mentioned in Checklist (limitations of our work), “we try to **avoid** theoretical analysis based on **impractical assumptions**. Therefore, some of our claims are supported by experiments and may require further theoretical investigation”. We do understand theoretical analysis is important to help us to understand underlying mechanism of the empirical results even with impractical assumptions on $\eta$ and $b$ (e.g. infinitesimal learning rate $\eta\rightarrow 0$, $b\ll n$ or $b=n$), but the community also needs analysis based on practical empirical settings.
> - Learning rate scaling only supported by a vague argument and empirical evidence.
>     - Previous scaling rules (LSR, SRSR) are based on wrong assumptions and thus does not consistent with empirical results (Fig 8 of Shallue et al. [42] and Fig 7 of Zhang et al. [57]). Our argument is not perfect but better explains the empirical results than previous ones.
>
> - My feeling is that the current theoretical setup may explain the gap between batch optimization and minibatch optimization but that is not the same as explaining generalization because generalization depends on the model class size. Can you agree or disagree?
>     - We agree with you that generalization depends on the model class size. Our work cannot fully answer how the model class size affects the generalization.
>     - However, lr ($\eta$) and batch size ($b$) in SGD play an important role in imposing the constraint on the interaction (IIR), which may be too strong for a small model, but appropriate for an over-parameterized model. Note that if $\rho_b/\eta$ is too large (upper left in Figure 5 heatmap) or too small (lower right in Figure 5 heatmap), i.e., the regularization effect is too strong or too weak, then the model does not generalize well.
>     - Most deep neural networks are over-parameterized ($N\ll m$, i.e., # training data $\ll$ # parameters) and the loss minimization for an over-parameterized model is an under-determined problem with multiple global minima (manifold of global minima). Some of these global minima generalize horribly and thus the model class size itself can not explain generalization.

---

> ### Author Response · Authors · 2022-08-01
> **A1-2. About Neural Networks**
>
> - Nothing in the theory related to neural networks.
>     - It does **not have to be a neural network**, but SGD with ($b,\eta$) and over-parameterization (large hypothesis space) matter. Inductive bias of the architecture of neural networks is not our focus in this paper.
> - More specifically, do I understand correctly that the theoretical setup does not assume anywhere that the model is a neural network? Then why are all the experiments on neural networks?
>     - It is not required to be a neural network, but as mentioned in A1-1, to generalize well under IIR, it is required to have a large hypothesis set, i.e., over-parameterized model. All experiments in Cohen et al. [6] (EOS) are on neural networks, but their theoretical setup also does not assume that the model is a neural network. We believe that general theoretical settings (rather than assuming a neural network) is rather a strength, not a weakness of our paper.

---

> ### Author Response · Authors · 2022-08-01
> **A1-3. About Linear and Saturation Scaling Rule (LSSR)**
>
> - Can we motivate the linear and saturation scaling rule better? I am not sure why keeping the right-hand side of Eq. (11) constant leads to optimal scaling.
>     - Keeping RHS of Eq. (11) constant $\frac{2\rho_b}{\eta}=C$ leads to **the same regularization effect** (IIR). Therefore, if $(b_1,\eta_1)$ and $(b_2,\eta_2)$ have the same $\frac{2\rho_b}{\eta}=C$, then they are expected to have **a similar generalization performance** as we have shown in Figure 5. We still do not know which $C$ value leads to the optimal generalization performance and the optimal $C$ value depends on data and the choice of neural network architecture.
> - Weak connection between the concepts proposed in the paper and generalization.
>     - There are many studies the sharpness is highly related with generalization (especially, Keskar et al. [23], Jiang et al. [22]). To be specific, the smaller the sharpness, the better the generalization. While we cannot reveal why this is the case in this paper, we can argue that (i) $(b,\eta)$ plays a crucial role in SGD dynamics and (ii) $(b_1,\eta_1)$ and $(b_2,\eta_2)$ impose the same implicit regularization on the model if they satisfies $2\rho_{b_1}/\eta_1 = 2\rho_{b_2}/\eta_2=C$.

---

### Author Response · Authors · 2022-08-01
**Dear reviewers**

Thanks for the valuable comments, suggestions and efforts towards improving our manuscript. Please kindly check the responses below.

---

> ### Author Response · Authors · 2022-08-09
> **Near the end of discussion period**
>
> We hope to discuss more about the reviews, but unfortunately there is no enough time for discussion. From now on, we are sorry that we may not answer the follow-up reviews.

---

### Public Comment · Authors · 2023-01-26
**Revised version will be published in ICLR 2023**

We thank the reviewers again. With their insightful and valuable comments, the contents and the clarity of our paper are much improved in the revised version. Please check our published version at the following link:
https://openreview.net/forum?id=bH-kCY6LdKg

---

### Meta-Review · Area_Chair_BXvz · 2022-08-28

**Recommendation:** Reject
**Confidence:** Certain

**Metareview:**

While this paper presents a series of interesting observation, e.g., self regularization around the edge of stability and learning rate scaling, in my view it fails to communicate the scientific value of the work in a coherent way. For example, I find it puzzling that the main result of the paper "implicitly regularized interaction" is stated as a definition (Def. 4) instead of a theorem, proposition, or a hypothesis. During the discussion, I realized that the authors use the word "regularization" in a slightly non-standard way. My questions to the authors are: (1) what does it mean to regularize the interaction? and (2) how does it relate to generalization?

**Award:**

No

---

### Decision · Program_Chairs · 2022-09-14

Reject